# Modeling state-dependent communication between brain regions with switching nonlinear dynamical systems

**Orren Karniol-Tambour, David M. Zoltowski, E. Mika Diamanti, Lucas Pinto[†],**
**Carlos D. Brody, David W. Tank, Jonathan W. Pillow**
Princeton Neuroscience Institute, [†]Northwestern University
{orrenkt,dzoltowski,mdiamanti,cbrody,dwtank,jpilllow}@princeton.edu
lucas.pinto@northwestern.edu

## Abstract

Understanding how multiple brain regions interact to produce behavior is a major challenge in systems neuroscience, with many regions causally implicated in common tasks such as sensory processing and decision-making. Moreover, neural dynamics are nonlinear and non-stationary, exhibiting switches both within and across trials. Here we propose multi-region switching dynamical systems (MR-SDS), a switching nonlinear state space model that decomposes multi-region neural dynamics into local and cross-region components. MR-SDS includes directed interactions between brain regions, allowing for estimation of state-dependent communication signals and sensory inputs effects. We show that our model accurately recovers latent trajectories, vector fields underlying switching nonlinear dynamics, and cross-region communication profiles in three simulations. We then apply our method to two large-scale, multi-region neural datasets involving mouse decision-making. The first includes hundreds of neurons per region, recorded simultaneously at single-cell-resolution across 3 distal cortical regions. The second is a mesoscale widefield dataset of 8 adjacent cortical regions imaged across both hemispheres. On these multi-region datasets, MR-SDS outperforms existing models, including multi-region recurrent switching linear models, and reveals multiple distinct dynamical states and a rich set of cross-region communication profiles.

## 1 Introduction

Advances in neural recording techniques and large-scale electrophysiology (Sofroniew et al., 2016; Song et al., 2017; Liu et al., 2021) have transformed systems neuroscience over the past decade, providing simultaneous measurements of neural activity across regions at high temporal and spatial resolutions. Experiments using these technologies have shown that neural computation is highly distributed (Steinmetz et al., 2019; Cowley et al., 2020; Musall et al., 2018; Allen et al., 2019; Makino et al., 2017; Allen et al., 2017; Gilad et al., 2018). The need for complementary multiregion methods has led to a flurry of activity in this area (Semedo et al., 2014; Kohn et al., 2020; Semedo et al., 2019; Glaser et al., 2020; Perich & Rajan, 2020; Keeley et al., 2020a;b; Gokcen et al., 2024). In this work, we build upon previous contributions in multiregion models, switching nonlinear time-series models, and transformer-based models for neural data. We develop a probabilistic model that accounts for multiregion activity through latent, nonlinear dynamical systems that evolve and communicate in a directed fashion at each timestep. The approach models high dimensional multi-region observations as emissions from coupled, low dimensional dynamical systems with explicit local dynamics and communication. We use a Transformer encoder with per-region embeddings for posterior inference. We additionally introduce a measure of the volume of communications between brain regions in the model, allowing us to quantify the directional 'messages' communicated between regions at each timepoint. We show information about stimuli and internal cognitive variables can be decoded from messages,

providing a description of information flow across brain regions. Finally, we apply our model to three simulations and two state-of-the-art multiregion neural datasets.

## 2 BACKGROUND

### 2.1 NONLINEAR NEURAL DYNAMICS

A wide variety of nonlinear dynamics models for sequential neural data have been introduced in recent years (Linderman et al., 2017; Pandarinath et al., 2018; Duncker et al., 2019; Kim et al., 2021). One popular approach is to model nonlinear dynamics using switching linear dynamical systems (Linderman et al., 2017; Nassar et al., 2019; Zoltowski et al., 2020). While switching linear models can accurately approximate nonlinear dynamics, they require multiple discrete states to instantiate a single nonlinear vector field, which reduces interpretability. Similarly, the expressivity of each linear regime depends on the number of latent dimensions, which can make visualization challenging. A separate approach is LFADS, which uses a sequential autoencoder with a generative recurrent neural network (RNN) Pandarinath et al. (2018). While this approach is powerful, the RNN typically must be high-dimensional to learn complex dynamics, which makes visualization of the full dynamics challenging. Additionally, LFADS instantiates a single set of dynamics, whereas our aim is to learn multiple nonlinear dynamical regimes. Kim et al. (2021) introduced PLNDE, a powerful approach which uses a neural-ODE to model nonlinear, low-dimensional dynamics. Our proposed method extends this approach in discrete time to allow for multiple regions and switching between multiple nonlinear dynamics flow fields. Finally, Ye & Pandarinath (2021) used a Transformer to fit a highly expressive nonlinear model of neural activity without explicit dynamics. Our approach similarly exploits the expressivity and trainability of Transformers, but constrains their use to an encoder, allowing us to efficiently perform inference.

### 2.2 SWITCHING NONLINEAR DYNAMICS

Two recent papers introduced switching nonlinear dynamics models for time-series data. Dong et al. (2020) proposed a model with switching nonlinear dynamics that uses an RNN inference network to infer continuous latents, with a collapsed variational inference approach for the discrete latent variables. Ansari et al. (2021) extended this approach to explicitly model discrete state duration through . We borrow Dong et al.'s approach to discrete state inference in the mean-field inference method proposed for MR-SDS.

### 2.3 MULTI-REGION MODELING AND COMMUNICATION

Various approaches to modeling multi-region neural data have been proposed (Semedo et al., 2014; Kohn et al., 2020; Semedo et al., 2019; Glaser et al., 2020; Perich et al., 2020; Perich & Rajan, 2020; Keeley et al., 2020a;b; Gokcen et al., 2024). Our model directly relates to mp-rSLDS, the multiregion recurrent switching LDS model introduced in Glaser et al. (2020). In this setup, the observation process is constrained such that continuous latent states map uniquely to observations originating in a particular brain region. MR-SDS can be seen as a generalization of mp-rSLDS with nonlinear dynamics, communication, and emissions.

### 2.4 DYNAMICS OF DECISION-MAKING

Many tasks studied by neuroscientists involve decision-making. A common paradigm is sensory evidence accumulation, in which an animal accumulates competing sensory cues towards a decision (Brunton et al., 2013; Mazurek et al., 2003). Previous work has focused on modeling dynamics of neural activity on single decision making trials, typically in individual brain regions (Latimer et al., 2015; Zoltowski et al., 2020; DePasquale et al., 2021; Luo et al., 2023). Two emerging experimental results from the decision making literature add complexity to modeling brain-wide neural dynamics. The first is that the neural basis of decision making appears to be highly distributed. Recent work shows evidence and decision related information is present in many brain regions (Pinto et al., 2020; 2019), and precise inactivation studies provide evidence that multiple regions are causally involved in accumulation and decision (Pinto et al., 2020). However, the precise role of different regions remains an open problem, underscoring the need for novel multiregion analysis methods that can address questions about cross-region communication. The second challenge is that decision-making behavior appears to exhibit persistent states in which different decision strategies are employed (Ashwood et al., 2020; Stone et al., 2020). This indicates neural

dynamics may be non-stationary within and across trials, motivating our use of switching nonlinear dynamics. Finally, cue-locked responses in early visual and intermediate brain regions are modulated by recent sensory stimuli and evidence history (Koay et al., 2019). These findings motivate our use of nonlinear interactions between brain regions, history dependence, and stimuli effects.

## 3 MULTIREGION, SWITCHING DYNAMICAL MODEL

### 3.1 SWITCHING NONLINEAR DYNAMICS AND COMMUNICATION WITH MULTIPLE REGIONS

Here we describe the MR-SDS model in detail (Figure 1). The model is designed to capture the following key features. First, our goal is to learn low-dimensional, nonlinear dynamics and corresponding latent trajectories underlying neural data. We expect that neural dynamics are nonstationary, and that different nonlinear vector fields may better describe neural dynamics at different timepoints. We therefore incorporate a discrete state that enables switching between different sets of nonlinear dynamics. Second, we are interested in modeling multiregion neural data, and thus our model is designed to separate the activity of each brain region into different continuous latent dimensions, such that within-region and across-region dynamics and communication are accessible for analysis. Finally, we include a nonlinear emission mapping between continuous latent and observed neural data, to reflect the nonlinear observation process of calcium imaging.

### 3.2 DISCRETE SWITCHES BETWEEN DIFFERENT DYNAMICAL REGIMES

The MR-SDS model has both continuous and discrete latent variables, respectively denoted by $\mathbf{x}$ and $z$. The generative process is

$$z_t \mid \mathbf{x}_{t-1}, z_{t-1} \sim \text{Cat}(\pi_t), \quad \pi_t = \text{softmax}(f_z(\mathbf{x}_{t-1}, z_{t-1})) \tag{1}$$

$$\mathbf{x}_t \mid \mathbf{x}_{t-1}, z_t, u_t \sim \mathcal{N}(f_{\mathbf{x}}^{z_t}(\mathbf{x}_{t-1}, u_t), Q^{z_t}) \tag{2}$$

$$\mathbf{y}_t \mid \mathbf{x}_t \sim \mathcal{N}(g(\mathbf{x}_t), R). \tag{3}$$

Above, $\mathbf{y}_{1:T}$ and $u_{1:T}$ are the observation and input sequences. The discrete state $z_t$ switches probabilistically at each timestep as a function of the continuous latent $\mathbf{x}$, as well as its own history (similar to a Hidden Markov Model or HMM). It is also possible for the inputs $u_t$ to effect transitions directly, as they do in our first simulation (4.1). The discrete state $z_t$ indexes the dynamical regime active at time $t$, with different dynamics $f_{\mathbf{x}}^{z_t}$ governing the continuous latent variable $\mathbf{x}_t$ in each discrete state. Hence, changes in the discrete state over time cause switches in the nonlinear dynamics. Notably, the model does not explicitly specify a condition or distribution on switching locations or times induced by the discrete transition dynamics *a priori*; transitions are probabilistic and not known in advance. Thus, the model is free to learn transitions driven by a combination of continuous and discrete states, and possibly external inputs.

### 3.3 MULTIREGION DYNAMICS AND EMISSIONS STRUCTURE

Importantly, we constrain the model to have multi-region structure (Figure 1). We consider a decomposition of the global continuous state into $K$ variables private to each region $\mathbf{x}_t = \{x_t^k\}_{k=1:K}$, and define the overall global dynamics function $f_{\mathbf{x}}^{z_t}$ using additive components acting on each region's latents $x_t^k$. Thus, the dynamics function for the $k$'th region is

$$x_t^k = f_{kk}^{z_t}(x_{t-1}^k) + \sum_{j \neq k} f_{kj}^{z_t}(x_{t-1}^j) + f_{ku}^{z_t}(u_t) + \epsilon_t^k \;\;,\;\; \epsilon_t^k \sim \mathcal{N}(0,\ Q_k^{z_t}) \tag{4}$$

We similarly decompose the observations as $\mathbf{y}_t = \{y_t^k\}_{k=1:K}$. Thus, the corresponding emissions function $g$ for the $k$'th region is

$$y_t^k = g_k(x_t^k) + \delta_t^k \;\;,\;\; \delta_t^k \sim \mathcal{N}(0,\ R_k) \tag{5}$$

Above, $f_z$, $f_{kj}^{z_t}$, $f_{kk}^{z_t}$, $f_{ku}^{z_t}$, and $g_k$ are nonlinear functions parameterized by neural networks. The generative model accounts for each region's observations $y^k$ as nonlinear 'private' emissions from a corresponding continuous latent variable $x^k$. The discrete state $z_t$ remains a global switch that selects which of $M$ local dynamics and communication functions are active at each timepoint. Thus, $f_{kk}^{z_t}$ indicates the nonlinear local dynamics in region $k$ active

at time $t$ in discrete state $z_t$. In practice, a single conditional dynamics subnetwork can be used for each $f_{kk}, f_{kj}, f_{ku}$ across discrete states, with the state index provided to the network at each layer. We note that emissions in the model are constrained such that latents from one region only map to that region's observations. MR-SDS can thus be seen as a generalization of the LDS based multiregion model, mp-rSLDS (Glaser et al., 2020). For spiking data, it's also possible to use the emissions mapping $y_t^k \sim Pois(g_k(x_t^k))$.

### 3.4 Estimating cross-region communication, or 'messages' and input effects

By explicitly inferring each region's local dynamics, communication and input processing functions, the model exposes estimates of communication streams between regions. We define messages and regional input effects in the model as follows:

$$m_t^{kk} = E_{q(z_t, x_{t-1}^k | y)}\left[ f_{kk}^{z_t}(x_{t-1}^k) - x_{t-1}^k \right] \tag{6}$$

$$m_t^{kj} = E_{q(z_t, x_{t-1}^j | y)}\left[ f_{kj}^{z_t}(x_{t-1}^j) \right] \tag{7}$$

$$m_t^{ku} = E_{q(z_t | y)}\left[ f_{ku}^{z_t}(u_t) \right] \tag{8}$$

where above the expectations are taken with respect to the approximate posterior over the latent states, $z$ and $x$. Notably, these messages combine additively to the estimated overall 'flow' of the $k$'th region's trajectory at each time point, thus decomposing the relative contribution of each region and of inputs. Thus, the model allows us to produce estimates and quantify communication and input effects for further downstream analysis. Furthermore, we can perform decoding analyses on messages to track how information about stimuli or internal cognitive variables flow across regions over time.

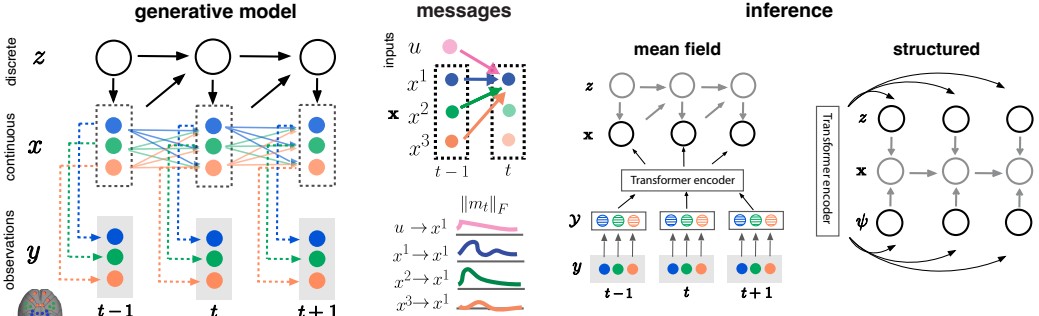

Figure 1: Generative model, messages and inference network for MR-SDS.

### 3.5 Amortized VI with multiregion structured Transformer network

We propose two variational inference (VI) schemes for MR-SDS: mean-field and structured. Both rely on a Transformer encoder with multiregion embeddings. In **mean-field**, the encoder directly produces independent means and covariances of the approximate posterior on **x**:

$$\mathcal{Y}_{1:T}^k = g_{emb}^k(y_{1:T}^k, \ u_{1:T}) \tag{9}$$

$$\mu_{1:T}^x, \Sigma_{1:T}^x = Transformer(\{\mathcal{Y}_{1:T}^k\}_{k=1:K}, \ u_{1:T}) \tag{10}$$

$$q_\phi(\mathbf{x}_{1:T} | \mathbf{y}_{1:T}, u_{1:T}) = \prod_{t=1}^{T} \mathcal{N}\big(\mathbf{x}_t \ ; \ \mu_t^x, \Sigma_t^x\big) \tag{11}$$

Above, $g_{emb}^k$ is a feed-foward neural network embedding region $k$'s activity $y_{1:T}^k$ along with observed stimulus inputs $u_{1:T}$. We use an encoder-only, non-causal Transformer architecture, motivated by the need to approximate the smoothing distribution of the continuous latents. Following (Dong et al., 2020), we treat samples $\hat{\mathbf{x}}_{1:T}$ from the approximate posterior as pseudo-observations for the subgraph $p(z_{1:T} | \hat{\mathbf{x}}_{1:T}, u_{1:T})$, which forms an HMM with $z$ as discrete states and $\hat{\mathbf{x}}$ as emissions. We perform conditionally exact inference for the discrete states $z$ with the Forward-Backward algorithm for HMMs, which we differentiate through

(Rabiner, 1989; Dong et al., 2020). With $\phi$, $\theta$ the parameters of the inference network and generative model, this results in a factorized variational approximation to the full posterior:

$$q(\mathbf{x}_{1:T}, z_{1:T}|\mathbf{y}_{1:T}, u_{1:T}) = q_\phi(\mathbf{x}_{1:T}|\mathbf{y}_{1:T}, u_{1:T})p_\theta(z_{1:T}|\mathbf{y}_{1:T}, \mathbf{x}_{1:T}, u_{1:T}) \tag{12}$$

A disadvantage of this approach is that the encoder must implicitly learn dynamics on $\mathbf{x}$ without using the generative model. In the **structured** approach, the Transformer encoder instead directly outputs an approximate posterior on $z$, as well as smoothing potentials $\psi$:

$$p_{1:T}^z, \mu_{1:T}^\psi, \Sigma_{1:T}^\psi = Transformer(\{\mathcal{Y}_{1:T}^k\}_{k=1:K}, \ u_{1:T}) \tag{13}$$

In this case the factorized variational approximation is:

$$q(\mathbf{x}_{1:T}, z_{1:T}|\mathbf{y}_{1:T}, u_{1:T}) = \prod_{t=1}^{T} p_\theta(\mathbf{x}_t|\mathbf{x}_{t-1}, u_t, z_t, \psi_t)q_\phi(z_t, \psi_t|\mathbf{y}_{1:T}, u_{1:T}) \tag{14}$$

The forward generative dynamics are conditioned on the discrete state $z$ and smoothing potentials $\psi$ to form the posterior on $\mathbf{x}$, thus directly reusing the generative model, as in structured VAEs (Archer et al., 2015; Johnson et al., 2016; Zhao & Linderman, 2023). While both schemes lead to good inference, We find the structured approach performs better generative prediction on high-dimensional neural datasets. In both inference approaches, the parameters of the generative model and inference network are learned jointly by maximizing the evidence lower bound (ELBO):

$$\mathcal{L}_{ELBO} = E_{q_\phi(\mathbf{x}_{1:T}|\mathbf{y}_{1:T}, u_{1:T})}\left[log\frac{p_\theta(\mathbf{y}_{1:T}|\mathbf{x}_{1:T})p_\theta(\mathbf{x}_{1:T}|u_{1:T})}{q_\phi(\mathbf{x}_{1:T}|\mathbf{y}_{1:T}, u_{1:T})}\right] \tag{15}$$

To avoid posterior collapse (Bowman et al., 2015), in which training produces good inference and reconstruction but poor generative dynamics, we utilize beta upweighting and 'overshooting' (Hafner et al., 2019). Briefly, in overshooting, we evaluate the likelihood of the latents and emissions by marginalizing over a multi-step predictive distribution:

$$\mathbf{x}_{1:T}^1 = E_{q(\mathbf{x}_{1:T}, z_{1:T}|\mathbf{y}_{1:T}, u_{1:T})}\left[f_\mathbf{x}^z(\mathbf{x}_{1:T})\right] \tag{16}$$

$$p_\theta(\mathbf{x}_{1:T}|u_{1:T}) = \beta \prod_{\tau=1}^{T} w_\tau^x p_\theta(f_\mathbf{x}^z(\mathbf{x}_{1:T}^{\tau-1})|u_{1:T}) \tag{17}$$

$$p_\theta(\mathbf{y}_{1:T}|\mathbf{x}_{1:T}) = \prod_{\tau=0}^{T} w_\tau^y p_\theta(\mathbf{y}_{1:T}|\mathbf{x}_{1:T}^\tau) \tag{18}$$

Above, $\mathbf{x}_{1:T}^\tau$ is the trajectory produced by applying dynamics $f_\mathbf{x}$ to the inferred latents $\tau$ times. This modified objective, similar to (Lusch et al., 2018), forces the emission distribution to pass through multiple timesteps of the dynamics, thus coupling the generative dynamics and emissions networks during training. It also encourages alignment of inferred latents and generative dynamics over multiple timesteps. The resulting multi-step predictive or 'overshooting' ELBO can be seen as a lower bound on the standard ELBO, which is recovered by setting T $= 0$ (Hafner et al., 2019). $w_\tau^x, w_\tau^y$ are weights that trade off inference ($\tau = 0$) and short vs. long term generative predictive accuracy. Finally, $\beta$ is a weighting factor, analogous to the $\beta$-VAE (Higgins et al., 2017), annealed over training. Setting $\beta > 1$ pushes the model to prioritize accurate dynamics over reconstruction.

## 4 Experiments

### 4.1 Multi-region switching Lotka-Volterra simulation

We first demonstrate the MR-SDS modeling approach in a multi-region simulation that switches between two sets of latent nonlinear Lotka-Volterra (LV) dynamics. Conceptually, our intention is to emphasize how learning switching nonlinear dynamics vector fields can uncover interpretable changes in dynamics over time. LV dynamics are a 2d model of interacting predator-prey populations (Lotka, 1925); LV models with regime switching have been used to model nonstationary environments and studied in the control theory literature (Li et al., 2009; Liu et al., 2013). These dynamics provide an example of non-stationary oscillatory coupling, a potentially important mode of neural communication (Kohn et al., 2020; Kastner et al., 2020; Modi et al., 2023; Khodagholy et al., 2017; Tal et al., 2020). We model each latent dimension as a separate region, with half of observations a function of $x_1$

and half a function of $x_2$. The simulation switches between two sets of LV dynamics, with the discrete switch driven by an external input $u_t$, representing a stimulus or upstream brain region: $p(z_{t+h}|z_t, u_{t+h}) \sim \text{Cat}(\text{softmax}(R_{z_t} u_{t+h}))$. The dynamics are:

$$\dot{x}_t^{(1)} = \alpha_{z_t} x_t^{(1)} - \beta_{z_t} x_t^{(1)} x_t^{(2)} \tag{19}$$

$$\dot{x}_t^{(2)} = \delta_{z_t} x_t^{(1)} x_t^{(2)} - \gamma_{z_t} x_t^{(2)} \tag{20}$$

Above, each parameter $\{\alpha, \beta, \delta, \gamma\}$ depends on the current discrete state $z_t$; each state corresponds to a different set of dynamics. We used the values $\{0.67, 1.33, 1, 1\}, \{0.9, 1.1, 1.2, 0.8\}$. Figure 2a shows dynamics for two states on both dimensions of the joint state space. $u_t$ is an external input and has value 1 when a switch is signaled and 0 otherwise, with switch probabilities controlled by the matrix $R$. (Figure 2). The model was simulated in discrete time using the Euler approximation. To ensure stability, we simulated 75,000 steps with a step size of 0.001 and then downsampled trials to 150 time steps. For each trial, between 1-3 discrete state switching times were sampled uniformly between 10 and 140. The model had linear observations, a multi-region embedding network, and was fit with $M = 2$ discrete states, with a global set of nonlinear dynamics in each state $x_t = f^{z_t}(x_{t-1}) + \epsilon_t$, with $x_t = [x_t^{(1)}, x_t^{(2)}]$ and $\epsilon_t \sim \mathcal{N}(0, Q^{z_t})$. Learned dynamics under the model matched the true dynamics, as shown by similarity in the dynamics vector fields (Figure 2B). Additionally, generated latent states on simulated test trials closely matched the known ground truth (Figure 2A,D). Next, we computed communication profiles. We computed the within region dynamics as $f(x_t^{(1)}, 0.0)$ and across region dynamics as $f(x_t^{(1)}, x_t^{(2)}) - f(x_t^{(1)}, 0.0)$. The profiles are marginalized over the discrete state posteriors. The generated communication profiles matched the true profiles (Figure 2C).

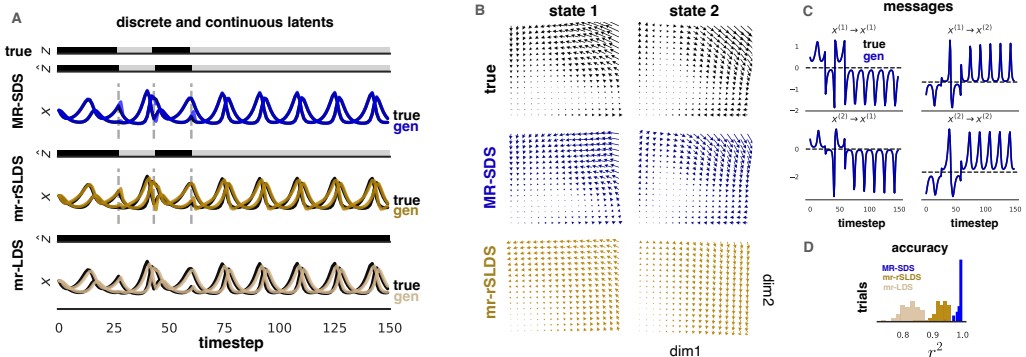

Figure 2: Multi-region Lotka-Volterra. **(A)** Example trial with comparison of true latents and latents generated by MR-SDS, mr-rSLDS, mr-LDS. Continuous latents were mapped via a linear transformation to account for non-identifiability. **(B)** True dynamics fields across two discrete states vs. dynamics fields learned by MR-SDS, mr-rSLDS. **(C)**. Overlaid true and generated messages between regions (corr. coef. 0.99). **(D)** Distribution of generated continuous latent $r^2$ values per trial for MR-SDS, mr-rSLDS, mr-LDS.

## 4.2 Multiregion evidence accumulation and reset with double-well dynamics

To further motivate the use of our model for neural data, we simulated a multiregion system performing a decision-making task with switching nonlinear dynamics (Figure 3). Animals performing evidence accumulation in sequential trials appear to display robust and structured sequences of behavior, including long periods of lapses, bias and strategy switches (Ashwood et al., 2020; Stone et al., 2020; Koay et al., 2019). Thus, our aim was to show how intrinsic dynamics of a hierachical, multiregion, switching nonlinear model can account for accumulation dynamics across consecutive trials in a way that produces history effects and bias; and to show how these dynamics could be recovered and analyzed with our approach. The simulated system was composed of two regions, an 'accumulator' and a 'controller', each with two distinct dynamical phases: accumulation and return. The accumulation dynamics of the accumulator were inspired by a classic model of perceptual decision making with

bi-stable attractor dynamics (Wong & Wang, 2006). During accumulation, the accumulator is presented with competing left and right evidence stimuli, and follows nonlinear dynamics leading to one of two attractor wells. The controller receives input from a single dimension of the accumulator, providing information about the progression of the accumulation process, but not the winning side. This drives the controller to its own attractor well; when it reaches this attractor, the return phase on both regions is triggered. During the return phase, the controller provides feedback to the accumulator, which follows a different set of nonlinear dynamics, returning it (roughly) to its initial condition. This final location serves as the initial condition for the next trial. The variability in resulting initial conditions leads to biased trial sequences, similar to those seen in animals (Ashwood et al., 2020; Stone et al., 2020). We provide further details in the appendix. MR-SDS recovered the correct discrete and continuous states, as well as the dynamics gradient fields for both regions and states. As an additional check on the quality of learned dynamics, we used the dynamics model to simulate new trajectories from the true initial condition and inputs, and found these were similar to true and inferred trajectories.

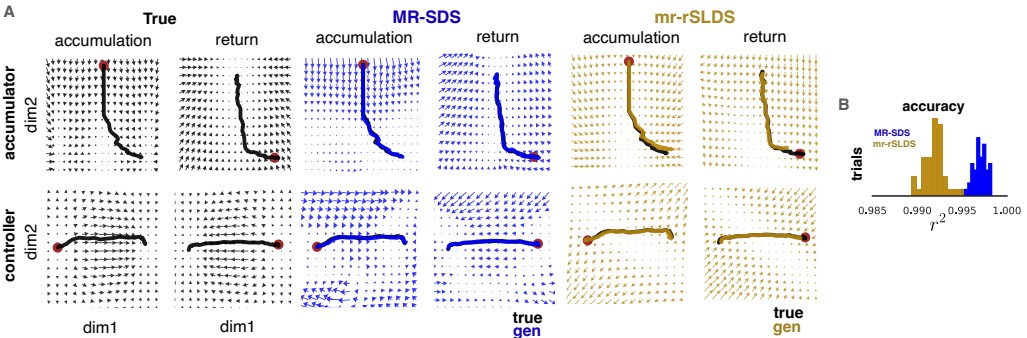

Figure 3: **(A)** True and generated latent trajectories and gradient fields for each region and 2 states for MR-SDS and mr-rSLDS. **(B)** Distribution of $r^2$ across trials .

### 4.3 Interacting high-dimensional RNNs driven by multiple inputs

A potential concern with state-space models amongst neuroscientists is that there is no physically instantiated low dimensional latent state in the brain — neural data comes from a high dimensional dynamical system with low-dimensional intrinsic dynamics. Thus, our intention in this simulation is to show that MR-SDS is able to recover important qualitative aspects of a multiregion system with high dimensional, nonlinear dynamics, despite model mismatch. Furthermore, we wish to show that our approach provides better interpretability than simply using PCA to visualize the true high-dimensional dynamics of the model. Following (Perich et al., 2020; Perich & Rajan, 2020), we simulated a multiregion system with 3 interacting, high-D (N=1000) RNNs, with sparse connectivity structure (Figure 4). Two RNNs were driven by separate inputs: a discrete step and a smooth sequence; the third RNN was driven solely by the other RNNs and exhibited chaotic dynamics. We emphasize that in this simulation, there is a mismatch between the true generative model and MR-SDS. We thus compared dynamics and latents recovered by MR-SDS to the true dynamics and trajectories projected onto their first 2 PCs. We found MR-SDS embeds a richer representation of the high dimensional dynamics into 2d than PCA, as well recovers important features of the system dynamics and communication.

### 4.4 Application: 3 region, single-cell resolution mesoscope data

We applied our method to calcium imaging data recorded in mice performing a sensory evidence accumulation task (Figure 4). In the task, a headfixed mouse runs on a linear track in virtual reality while columns ('towers') are presented on both sides of the track (Pinto et al., 2019). The mouse must decide on which side more towers were presented and turn correctly to get a reward following a short memory delay section in the track. We analyzed a single day of mesoscope calcium imaging data of 3 distal brain regions in a single hemisphere, consisting of 178 correct trials. The regions were: AM (a visual area), retrosplenial cortex (RSC), and M2 (a higher level motor / planning area). Preliminary results showed distinct

trial averaged communication profiles for left and right trials, as well as strong stimuli effect for visual area AM.

| model | $K$ | $L$ | $D$ | regions | mesoscope | widefield |
|---|---|---|---|---|---|---|
| mr-PCA | 1 | 1 | 2/2 | 3/8 | 0.0045469 | 9.083e-05 |
| mr-LDS | 1 | 1 | 2/2 | 3/8 | 0.0043695 | 0.000133 |
| mr-ar-LDS | 1 | 2 | 3/2 | 3/8 | 0.0043951 | 0.0001337 |
| mr-SLDS | 2 | 1 | 2/2 | 3/8 | 0.0043723 | 0.0001506 |
| mr-rSLDS | 2 | 1 | 2/2 | 3/8 | 0.0043632 | 0.000157 |
| mr-ar-rSLDS | 2 | 2 | 3/2 | 3/8 | 0.0043988 | 0.0002 |
| mr-rSLDS | 2 | 1 | 5/- | 3/- | 0.0042142 | - |
| mr-rSLDS | 2 | 1 | 10/- | 3/- | 0.0040676 | - |
| **MR-SDS (biRNN)** | 2 | - | 2/2 | 3/8 | **0.0040761** | **7.444e-05** |
| **MR-SDS** | 2 | - | 2/2 | 3/8 | **0.0037734** | **7.491e-05** |
| lds | 1 | 1 | 9/24 | 1/1 | 0.0040886 | 0.0001144 |
| rslds | 2 | 1 | 9/24 | 1/1 | 0.0040877 | 0.00011744 |
| LFADS | 1 | - | 200/200 | 1/1 | 0.00434653 | 9.240e-05 |

Table 1: Held-out trial and neuron co-smoothing test MSE, across models on mesoscope and widefield datasets. Top: multiregion. Bottom: single region models. $K$: discrete states. $L$: time-lags. $D$: latent dimensions per region.

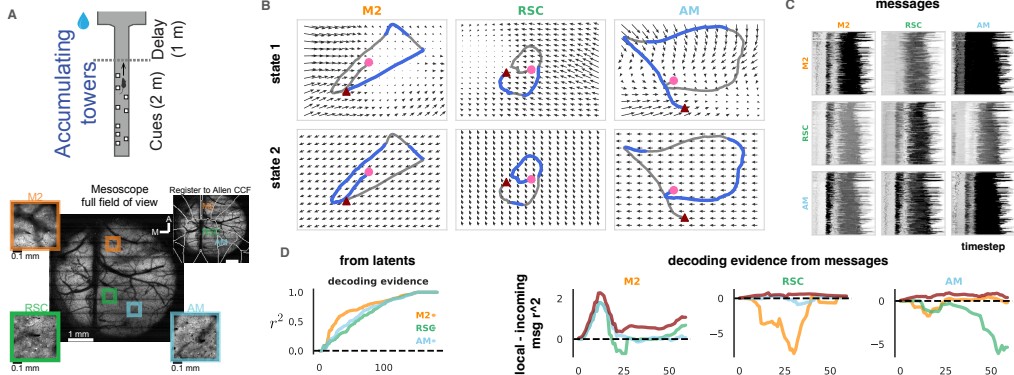

Figure 4: Mesoscope imaging data. **(A)** Towers task and multiregion imaging at single cell resolution. **(B)** Inferred mean latent trajectories and gradient fields for each region and 2 states; mean occupied state in blue. **(C)** Inferred message norms from sending (columns) to receiving (rows). Decoding evidence. Left: from latents. Right: relative information about evidence present in incoming messages compared to local dynamics. Incoming stimuli effect in red. Note big incoming spike in M2, explaining higher decodability from latents on left plot.

### 4.5 APPLICATION: 8 REGION, MESOSCALE WIDEFIELD DATA ACROSS HEMISPHERES

We also applied our method to wide-field calcium imaging data from the same experimental task. 16 regions (8 bilaterally) were imaged across the cortical surface (Figure 5). We analyzed a single day of recordings consisting of 63 correct trials. To analyze communication between regions, we examined estimated messages under the inferred model parameters and approximate posterior over continuous latents. To quantify the messages, we computed their Frobenius norm at each timepoint. Figure 5d shows a 'macro' picture of communication streams between all regions. Highlighted are streams with notable feedforward, feedback and stimuli components. We draw attention to a few important features. First, profiles showed strong feedforward drive from V1 to mV2 and PPC. Second, PPC showed strong feedback drive back to V1 and mV2, as well as feedforward drive to upstream regions RSC and SS. These results are potentially consistent with a hypothesized central role of PPC in evidence accumulation, suggesting PPC may act as a hub between early visual areas and upstream regions (Pinto et al., 2019). Notably, PPC also showed messages incoming with high information about evidence compared to local dynamics (Figure 5c). Further analysis

may seek to probe the timing of single trial communications, as well as their relationship to evidence levels in the trial.

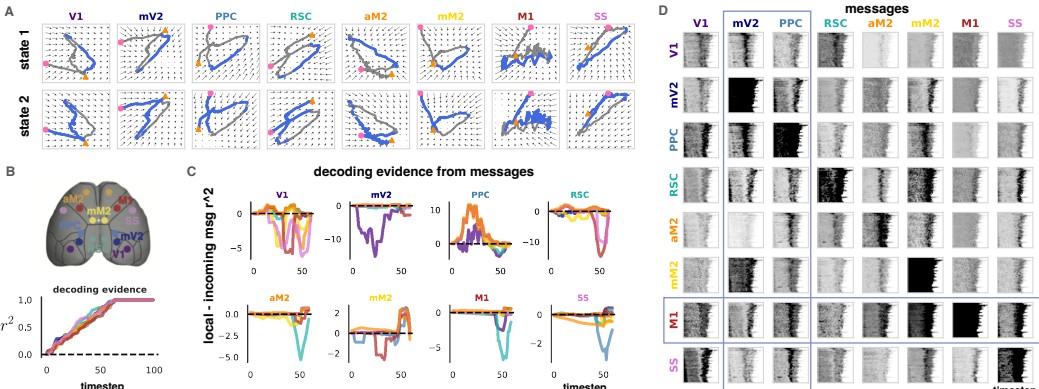

Figure 5: Mesoscale widefield imaging data. **(A)** Inferred mean latent trajectories and gradient fields for each region and 2 states; mean occupied state in blue. **(B)** Top: Imaging performed over 8 regions across both hemispheres. Bottom: decoding evidence from MR-SDS inferred latents. **(C)**. Relative information about evidence present in incoming messages vs. local dynamics. **(D)** Macro view of communication streams. Inferred message norms from sending (columns) to receiving (rows). We highlight upstream communication from mV2 and PPC to higher order areas, and M1 receiving broad inputs from all regions.

## 5 DISCUSSION

We propose a switching, nonlinear dynamics approach for modeling multi-region neural data, comparing it with piece-wise linear models, and show that our model compares favorably with these in simulations. We emphasize that our simulations were designed to capture important aspects of multiregion communication, rather than performance gains and expressivity provided by nonlinear dynamics and emissions. We therefore also included comparisons on calcium imaging neural data, showing MR-SDS more accurately predicts held out neurons' activity, reflecting higher expressivity and performance benefits. A key benefit of the model's nonlinear dynamics and emissions is the ability to represent rich nonlinear dynamics and account for a nonlinear emissions process. Piece-wise linear multiregion models, by contrast, require higher latent dimensions and additional discrete switches to reach similar expressivity; we argue this is a disadvantage because it requires use of dimensionality reduction methods to visualize the resulting dynamics. Similarly, while RNN based models such as LFADS are extremely powerful, they generate dynamics in a very high dimensional space, making dynamics challenging to visualize. While our approach uses a Transformer, which is challenging to interpret, we constrain its use to our inference network, which does not need to be interpreted for scientific analysis, and use simple feed-forward networks to parameterize generative dynamics in our model. An interpretational limitation of MR-SDS, as in any switching or clustering model, is the relative advantage of lumping vs splitting, given that switching nonlinear dynamics can also be described by a single set of nonlinear dynamics in a transformed space. There remain important aspects of future work. In particular, our method does not account for unobserved regions or neurons, and learned interactions in MR-SDS are not causal. It may also be desirable to learn the number of regions, or find clusters of neurons across regions with coherent functional dynamics. Additionally, future work should explore alternative types of multiregion communication, and demonstrate whether our approach succeeds or fails in each case.

## 6 ACKNOWLEDGEMENTS

We thank Brian DePasquale, Stephen Keeley, Josh Glaser and Matt Whiteway for helpful discussions.

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

## A  APPENDIX

### A.1  INTERACTING HIGH-DIMENSIONAL RNNS DRIVEN BY MULTIPLE INPUTS — ADDITIONAL DETAIL

Additional: The first RNN was externally driven by a sequence input representing a sequentially active neural population, and the second RNN was externally driven by a discrete step signal, representing a population generating fixed points. The third region was driven by both the first and second RNNs, and exhibited chaotic dynamics. The simulation in (Perich et al., 2020) was modified to make the determinics RNN dynamics stochastic and additive across nonlinear components.

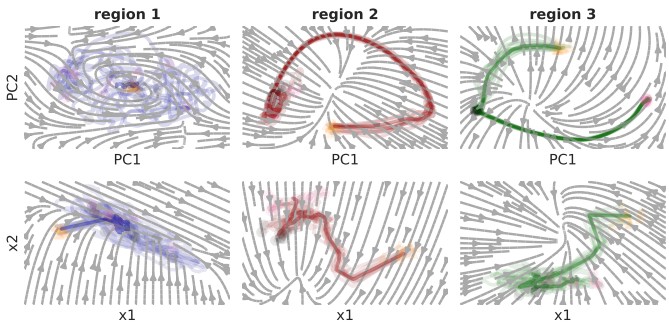

Figure 6: Interacting high-D RNNs driven by inputs. Placeholder fig

A.2 Hyperparameter sweeps.

| model | $K$ | $L$ | $D$ | regions | mesoscope | widefield |
|-------|-----|-----|-----|---------|-----------|-----------|
| **MR-SDS** | 1 | - | 2/2 | 3/8 | 0.00405374 | **8.335e-05** |
| **MR-SDS** | 2 | - | 2/2 | 3/8 | 0.00392798 | **8.267e-05** |
| **MR-SDS** | 3 | - | 2/2 | 3/8 | 0.00387180 | **-** |
| **MR-SDS** | 2 | - | 3/3 | 3/8 | 0.00390648 | **7.743e-05** |
| **MR-SDS** | 2 | - | 4/4 | 3/8 | 0.00387056 | **7.739e-05** |
| **MR-SDS** | 2 | - | 5/5 | 3/8 | 0.00389255 | **7.982e-05** |
| **MR-SDS** | 2 | - | -/2 | -/16 | **-** | **8.304e-05** |

Table 2: Held-out trial and neuron 1-step generative co-smoothing test mean square error, across hyperparameter sweeps of number of states and number of dimensions per region. Final row includes a 16 region result for the widefield dataset. $K$: number of discrete states. $D$: dimensions of latent dynamics used per region.

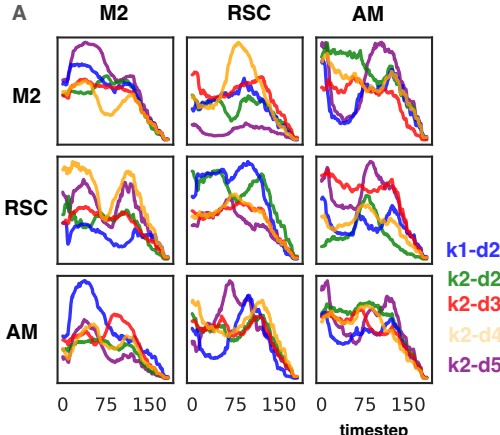

Figure 7: We compare the mean message norms across trials for each of the model hyper parameters on the mesoscope dataset.

All experiments and analysis were run on a 28 CPU, 8 GPU (GeForce RTX 2080 Ti) server.

A.3 Multiregion RNN simulation, additional detail.

A.4 Evaluating performance on simulation.

We note that inferred messages in MR-SDS are produced by combining inferred latents at each timestep with a single generated step through the dynamics model. Thus, when

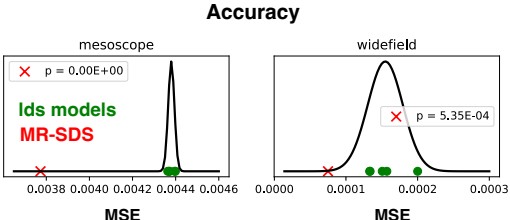

Figure 8: We compare the mean square error on the co-smoothing target for LDS based models (green) and MR-SDS (red). We compute p-values based on Gaussian fits to the other scores, which are highly significant for both datasets

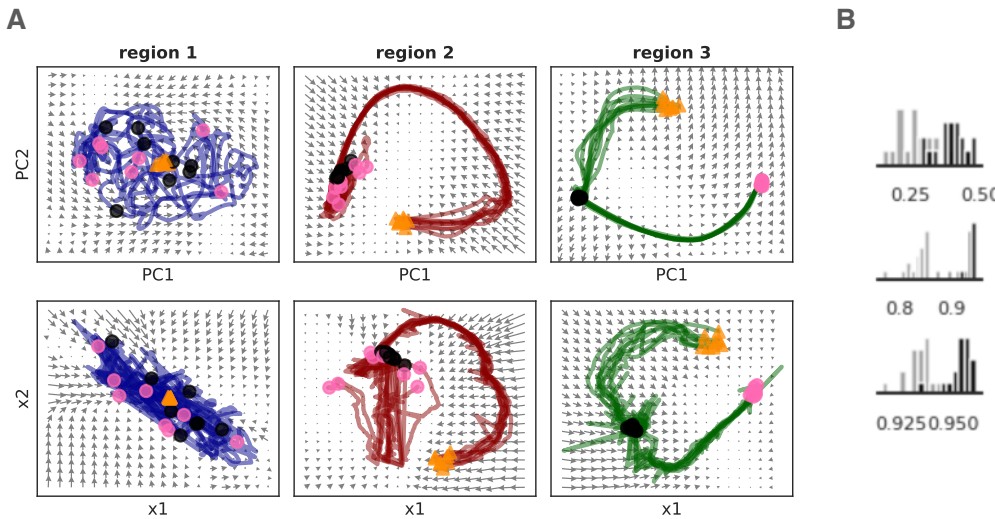

Figure 9: **(A)** Top: network activity projected onto first two PCs. Bottom: inferred latents using MR-SDS. **(B)** $r^2$ across trials per region, on reconstruction of observations from first 2 PCs vs under model.

validating MR-SDS on simulations, we took special care to evaluate the accuracy of single-step latent generation under the generative model, rather than inference alone. In single-step latent generation, we first infer the latents $\{\hat{x}_t\}_{1:T-1}$ under the inference model, and then evaluate $\{f(\hat{x}_t)\}_{1:T-1}$ where $f$ is the generative dynamics.

## A.5    EVALUATING PERFORMANCE ON NEURAL DATASETS

To evaluate the performance of on neural data, we compared held out model predictions for a family of related multi-region latent variable models (Table 2). These include PCA, LDS, SLDS, and rSLDS. Importantly, these models can be seen as special cases of MR-SDS, instantiated with less expressive components: PCA has no dynamics and linear emissions; rSLDS has linear transitions, dynamics and emissions. We used the 'co-smoothing' metric (Macke et al., 2011; Pei et al., 2021), which evaluates reconstruction loss on 25% held-out neurons in held-out trials. Cosmoothing requires infering latents at test time based on a subset of neurons, and predicting activity of a held out set from those latents. The latent representation thus has to be robust to input removal, a more challenging test than simple reconstruction. MR-SDS model better performance than comparable models on both datasets. We additionally compared to rSDLS models with higher latent dimensionality, observing that mr-rSLDS models required 10 latent dimensions per region (vs 3) to achieve comparable

performance on the mesoscope dataset. Finally, we found MR-SDS was comparable with rSLDS models fit to the entire dataset as a single region - providing a baseline for performance without multiregion description.

## A.6   Cosmoothing test time evaluation

A key question in evaluating unsupervised models of neural data is regarding the appropriate metric to use to assess model fits. A limitation of looking only at reconstruction error on held out data is that the model may simply learn to compress and decompress the observations, effectively learning an identity mapping. This can be seen in the relative strength of the PCA benchmark (more on this in MR-PCA section below) on test reconstruction alone. An alternative suggested by (Macke et al., 2011) and adopted in the recent Neural Latents Benchmark (Pei et al., 2021), is "co-smoothing", in which a model is evaluated on reconstruction loss of held-out neurons on held-out trials (see Figure A1). This tests the ability of the model to learn a latent representation of the data that is robust to removal of inputs, and is thus a more challenging test of a model's performance. A key result in the benchmarks is that models that perform reasonably well on test reconstruction may perform poorly on cosmoothing. The Neural Latents Benchmark uses a 25% neuron drop out rate, and we apply this rate, with one modification - we evaluate all models against a 25% drop out rate per region. This ensures that results are not dominated by regions with more neurons.

## A.7   Cosmoothing multiregion dropout training

To improve model generalization and to allow it to accurately infer latent variables given only a subset of neural responses, we trained the model on real neural data by dropping out a subset of inputs to the inference network in each batch. In particular, we dropped out inputs from individual neurons over time. Notably, it was important that we drop out equivalent fractions of neurons from each brain region.

To make MR-SDS and the latent represention it learns robust to missing inputs inherent in the co-smoothing test, we modify the training procedure as presented in algorithm 1.

There, the monte carlo samples used to evaluate the expectation are taken with respect to the approximate posterior evaluated on the dropped inputs. We dropped out 25% of neurons per region. We found that dropout of 50-80% of trials works well, depending on the dataset. Empirically, datasets with higher correlation between observations, such as the widefield dataset used in the experiments, require a lower trial dropout rate. We note that this is similar to the 'coordinated dropout', or 'speckled' holdout strategy used by (Keshtkaran & Pandarinath, 2019), but extended here to multiregion data, and we drop out all of the timepoints for random neurons on each trial.

We used a dropout rate $p_1 = \{0., 0.5\}$ for the mesoscope and widefield datasets respectively in the experiments presented in the paper.

---

**Algorithm**  Cosmoothing multiregion drop out training

---

**Inputs:** batch size $m$ , region sizes $\{r_1, ..., r_R\}$
trial dropout rate $p_1$, neuron dropout rate $p_2$

**for**  batch $b = \{Y_{1:k}^l\}_{l=1:m} \in \mathcal{D}_{train}$  **do**
    Create batch mask $\tilde{b} \leftarrow 1$
    Sample $|m| = p_1 m$ trials from $\tilde{b}$
    **for** region $r \in 1 : R$ **do**
        Sample $|n_r| = p_2 n_r$ neurons for each trial $l \in m$
        Set mask $\tilde{b}_{n_r}^m \leftarrow 0$
    **end for**
    $\tilde{Y} = Y \odot \tilde{b}$ (ie input 'dropout')
    Evaluate the elbo as:  $E_{q(Z,X|\tilde{Y})}[\log p(Z,X,Y)] + H[Z,X]$
    Take training step
**end for**

---

Figure 10: Multiregion cosmoothing and multiregion cosmoothing dropout training.
**Left:** When cosmoothing, we drop a percent of neurons from each region on each test trial and present this as inputs to the model; we then evaluate the reconstruction error on those neurons alone. **Right:** In multiregion cosmoothing dropout training, we drop a percent of neurons from each region on some percent of training trials, and evaluate reconstruction error on all neurons.

| dataset | $p_1$ | test | cosmooth |
|---------|-------|------|----------|
| mesoscope | 0.8 | 0.0040009 | 0.0040761 |
| mesoscope | 0 | 0.0040569 | 0.0045115 |

Table 3: Held-out test error with and without multiregion dropout training.

## A.8  Training details

## A.9  Double well simulation details

The simulation switches between two sets of dynamics, accumulation dynamics, and return dynamics. The system switches from accumulating to return at fixed time following the stimulus presentation and memory periods, mimicking the visual cue marking the end of the maze and the beginning of the maze choice arms for the animal in the real experiment.

$$\dot{x}_{t,accum}^{(1)} = a_{accum}^{(1)} \left( \begin{bmatrix} x_{t,0}^{(1)} - (x_{t,0}^{(1)})^3 \\ -x_{t,1}^{(1)} \end{bmatrix} + u_t + \sigma_t^{(1),acc} \right) \ , \ \sigma_t^{(1),acc} \sim \mathcal{N}(0, \Sigma_{acc}^{(1)}) \tag{21}$$

$$\dot{x}_{t,accum}^{(2)} = (A_{accum}^{(2)} - I)x_t^{(2)} + (1 - 2x_{t,1}^{(1)}) + \sigma_t^{(2),acc} \ , \ \sigma_t^{(2),acc} \sim \mathcal{N}(0, \Sigma_{accum}^{(2)}) \tag{22}$$

$$\dot{x}_{t,ret}^{(1)} = a_{ret}^{(1)} \left( -x_t^{(1)} + b_1 \right) + c^{(1)}x_{t,1}^{(1)} + \sigma_t^{(1),ret} \ , \ \sigma_t^{(1),ret} \sim \mathcal{N}(0, \Sigma_{ret}^{(1)}) \tag{23}$$

$$\dot{x}_{t,ret}^{(2)} = (A_{ret}^{(2)} - I)x_t^{(2)} + (1 - 2x_{t,1}^{(1)}) + \sigma_t^{(2),ret} \ , \ \sigma_t^{(2),ret} \sim \mathcal{N}(0, \Sigma_{ret}^{(2)}) \tag{24}$$

The following values are used for the variables:

$$a_{accum}^{(1)} = a_{ret}^{(1)} = 0.1 \tag{25}$$

$$b^{(1)} = [0, 0.5] \tag{26}$$

$$c^{(1)} = [0.005, 0.03] \tag{27}$$

$$\Sigma_{accum}^{(1)} = 0.0005I \tag{28}$$

$$\Sigma_{ret}^{(1)} = \begin{bmatrix} 0.035 & 0 \\ 0 & 0 \end{bmatrix} \tag{29}$$

$$A_{accum}^{(2)} = 0.5I \tag{30}$$

$$A_{ret}^{(2)} = 0.1I \tag{31}$$

$$\Sigma_{accum}^{(2)} = \Sigma_{ret}^{(2)} = 0.055I \tag{32}$$

$$\tag{33}$$

## A.10 BENCHMARKS

We provide details on the models used in the benchmarks presented across experiments.

### A.10.1 MR-PCA

PCA is a surprisingly powerful baseline on many unsupervised machine learning tasks, e.g. (Bojanowski et al., 2017). We include in the benchmarks a comparison to multi-region PCA (MR-PCA), by which we mean fitting PCA to each region's training data separately. Test data for each region is then reconstructed by first projecting it onto the top $d$ principal components for that region. We note that while MR-PCA is a strong baseline for test, it can only compress data linearly, and this performs poorly on co-smoothing on both datasets, emphasizing the importance of this metric. Additionally, we note that MR-PCA is a much stronger baseline on the widefield dataset, which is consistent with the low intrinsic dimensionality of the data (see Figure A3 of cumulative variance explained for both datasets).

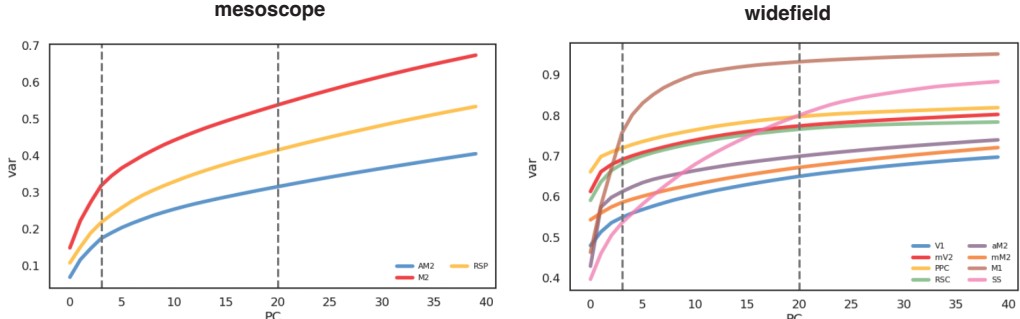

Figure 11: Variance explained by PC dimension per region, for both real datasets. Intrinsic (linear) dimensionality of the mesoscope data is higher than widefield.

### A.10.2 Multiregion LDS, SLDS, and rSLDS models

Briefly, (Glaser et al., 2020) introduced a multiregion rSLDS model of the following form:

$$z_t \sim \mathrm{Cat}(\pi_t) \;,\;\; \pi_t = \mathrm{softmax}\Big(\sum_k R^k_{z_{t-1}} x^k_{t-1} + r^k_{z_{t-1}}\Big) \tag{34}$$

$$x^k_{t+1} = A^{z_t}_{kk} x^k_t + \sum_{j \neq k} A^{z_t}_{kj} x^j_t + B^{z_t}_k u_t + \epsilon^k_t \;,\;\; \epsilon^k_t \sim \mathcal{N}(0, Q^{z_t}_k) \tag{35}$$

$$y^k_t = f(C_k x^k_t + d) \tag{36}$$

Where above $A, B, R, Q, C$ are matrices and $r, d$ are bias vectors. We refer to the model above as an MR-RSLDS. Similarly, we refer to versions of this model with 1 discrete state as an MR-LDS, and versions with no recurrence in the switching dynamics as MR-SLDS.

All LDS variant models were run in parallel on 28 CPUs using a modified version of the SSM package [Linderman et al https://github.com/ slinderman/ssm], using variational Laplace EM (vLEM) (Zoltowski et al., 2020) .

### A.10.3 Higher-order auto-regressive, or AR(p) MR-RSLDS models

We include in the benchmarks comparison LDS and rSLDS models with higher order auto-regressive continuous latent dynamics. These models (AR(p) MR-RSLDS) have the following modified latent dynamics:

$$x^k_{t+1} = \sum_{i=0}^{\tau} \Big[ A^{z_t,i}_{kk} x^k_{t-i} + \sum_{j \neq k} A^{z_t,i}_{kj} x^j_{t-i} + B^{z_t,i}_k u_{t-i} \Big] + \epsilon^k_t \;,\;\; \epsilon^k_t \sim \mathcal{N}(0, Q^{z_t}_k) \tag{37}$$

Above, $\tau$ is the order of the AR process, or the total number of lags, with $i$ indexing the $\tau$ lagged dynamics, communication, and inputs matrices, $A^{z_t,i}_{kk}$, $A^{z_t,i}_{kj}$, and $B^{z_t,i}_k$. Similarly, $x_{t-i}$ and $u_{t-i}$ represent the $i$'th lagged state and input at time $t$. The emissions and discrete latent transition dynamics remain as in the single lag models.

Adding higher order autoregressive dynamics adds to the expressivity of this class of models. While dynamics remain piecewise linear, they are no longer linear with respect to the previous timepoint alone. We note that these models performed better than single lag models (see results in Tables 1,A1).

In order to fit these models, we extended vLEM (Zoltowski et al., 2020) to higher order lags. Briefly, this involved modifying the Hessian to account for higher order terms.

A.10.4   Single region models

In the benchmarks, we include a comparison with single region LDS and rSLDS models that have a latent dimension equal to the sum of the latent dimensions used across regions in the multiregion models. Fitting a single region model to multiregion data results in a meaningful loss of interpretability, because there are no per-region latents, and no estimates of communication. However, these models in general can achieve lower test and cosmooth errors, since they are free to use latent dimensions to explain any part of the data across regions. As such, they give a baseline for performance on these datasets. Table 1 shows that MR-SDS achieves lower cosmoothing test error than both single region LDS and rSLDS models on both datasets, thus providing an increase in both performance and interpretability. Table 2 shows that a better calibrated model improves in test error as well relative to single region LDS and rSLDS models on mesoscope data, but not widefield data. We think this is likely due to the higher intrinsic dimensionality of the mesoscope data (see earlier comments on dimensionality in the MR-PCA section). x

A.11   Architecture and training details

We provide additional details on the network architecture and hyperparameters used to fit the model for each experiment:

| | $K$ | $R$ | $d$ | $f_{kk,kj}$ | $f_{ku}$ | $g_k$ | $g_{emb}$ | $g_{birnn}$ | $g_{rnn}$ | $g_x^{\mu,\Sigma}$ | $p_1$ | lr | steps |
|---|---|---|---|---|---|---|---|---|---|---|---|---|---|
| LV | 2 | 2 | 1 | 32x2 | - | L | - | 3;3 | 6 | 64,32 | 0 | 1e-3 | 2.5e3 |
| DW | 2 | 2 | 3 | 32x2 | L | L | 16 | 3;3 | 6 | 32x2 | 0 | 1e-3 | 5e3 |
| RNN | 2 | 2 | 3 | 32x2 | L | L | 16 | 3;3 | 6 | 32x2 | 0 | 1e-3 | 5e3 |
| Meso | 2 | 3 | 3 | 32x2 | 4x2 | 128x2 | 32x2 | 32;32 | 64 | 256 | 0.8 | 1e-3 | 15e3 |
| Wide | 2 | 8 | 2 | 32x2 | 4,2 | 32x3 | 32,16x2 | 32;32 | 64 | 128 | 0.5 | 2e-4 | 15e3 |

Table 4: Parameters used for each of the experiments presented in the paper. LV: Switching Lotka-Voltera. DW: Double well. Meso: mesoscope. Wide: widefield. lr: learning rate. $p_2$: trial dropout rate used for cosmoothing training. $I$: Identity mapping. L: linear layer. $d$: dimensionality of latents for each region. For Bi-RNN, semicolon (;) indicates a single layer with two concatenated RNNs of that size, one forward and one backward.

A.12   Explicit duration

In the explicit duration case, the full generative model is:

$$\rho_m(c) = \text{Cat}(d_{min}, ..., d_{max}) \tag{38}$$

$$v_m = 1 - \frac{\rho_m(c)}{\sum_{d=c}^{d_{max}} \rho_m(d)} \tag{39}$$

$$c_t \mid z_t, c_{t-1} \sim \begin{cases} v_{z_{t-1}}(c_{t-1}) & \text{if } c_t = c_{t-1} + 1 \\ 1 - v_{z_{t-1}}(c_{t-1}) & \text{if } c_t = 1 \end{cases} \tag{40}$$

$$z_t \mid \mathbf{x}_{t-1}, z_{t-1}, c_t \sim \begin{cases} \text{Cat}(\pi_t) & \text{if } c_t = 1 \\ \delta_{z_t = z_{t-1}} & \text{if } c_t > 1 \end{cases} \tag{41}$$

In words, each state $m$ has a learned categorical distribution $\rho_m(c)$ over durations. Counts at time $t$ are then sampled from a normalized version of this distribution, according to the discrete state $m$ active at time $t$ and denoted by $z_t$. Conditioning on the counts allows the discrete state to persist without sampling a new state through the transition distribution $\pi$. The choice of hyper parameters $d_{min}, d_{max}$ confers an inductive bias on the augmented model, that impacts the persistence of inferred discrete states.

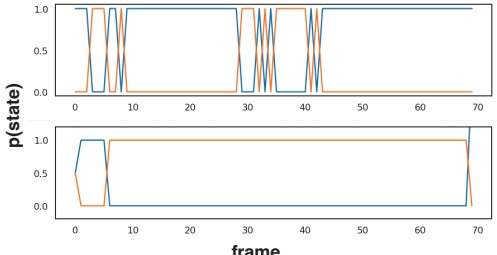

Figure 12: Explicit duration results in longer state persistence.
**Above** Inferred discrete state for single mesoscope trial with no learned explicit duration variabl. **Bottom:** Same, with explicit duration. Note the inferred discrete state persists for longer.

