# OpenReview forum: "Modeling state-dependent communication between brain regions with switching nonlinear dynamical systems"
_ICLR.cc/2024/Conference — ICLR 2024 poster_

### Official Review · Reviewer_LTdS · 2023-10-29

**Soundness:** 2 fair
**Presentation:** 2 fair
**Contribution:** 2 fair
**Rating:** 5
**Confidence:** 4

**Summary:**

This paper proposed the multi-region switching nonlinear state space model (MR-SDS) for learning state switching and nonlinear evolution of continuous latent with multi-region interactions setup. The inference algorithm uses the forward-backward algorithm to learn the latent state, and a transformer encoder as the variational distribution to infer the continuous latent. Experiments show that the proposed MR-SDS is able to learn the model parameter and then infer the nonlinear latent and predict future data. Besides, the volume of messages passing between different regions can be understood by such a model.

**Strengths:**

* The maths explaining the model is clear to me, with an intuitive schematic for the generative process.
* The model is able to capture the nonlinear evolution of the latent process $x_t$.

**Weaknesses:**

* The -2th line of page 3, $f_z^{kk}$ should be $f_{kk}^z$
* Then main weaknesses is the mismatches between texts and figures:
    * We know Fig. 1 is a summary of the whole generative model, but it is informal to not mention it in the main text.
    * Page 6, 2nd line should be Figure 2(c).
    * Page 6, 3-5 lines and Figure 2. Where is the $R$ and $u$ in Figure 2.
    * Figure 2(b), what is true, what is the dashed curve? There is no clear legend.
    * Other mismatches in Figure 2, and etc.
* No "." in the last sentence of section 4.1.
* I think forward Euler is not appropriate for generating the data. Although I don't understand different curves in Figure 2(b), I can see the numerical error amplifies significantly because of forward Euler.
* Why only one state switch is allowed in experiment 4.1 and has to be within 40-60? This generating process is too simple and not complicated enough to validate the model
* Overall, too many errors in this paper. I cannot understand most of the experiments since wrong text-figure assignments, no legend, etc.

**Questions:**

* Why ‘posterior collapse’ is enclosed single quotes, and also others.
* What about other methods for section 4.1 and 4.2? Is that other methods in Table 1 not applicable to experiment 4.1 and 4.2? For, example, the generative model of the Lokta-Voterra model is nonlinear. Authors show that MR-SDS is able to capture the nonlinear evolution of the then latent process $x_t$. But for SLDS, for example, $x_{t+1} \sim \mathcal N(Ax_t,\Sigma)$. It seems like $Ax_t$ is a linear process, but there is a Gaussian distribution, so it is actually a nonlinear model. No matter how we define "linear/nonlinear", why not test whether SLDS can learn the Lokta-Voterra latent?
* Is Table 1 the held-out likelihood? Where is the variance? Are the likelihood of MR-SDS significantly better than others? Is it likelihood, or log-likelihood?
* I cannot see the significant benefits of the proposed model than others, in terms of both prediction on future data and latent estimation.

---

> ### Author Response · Authors · 2023-11-21
>
> We thank the reviewer for their time, comments and helpful suggestions. We especially appreciate the reviewer’s positive comments on the clarity and intuitive presentation of the model and approach in our paper, as well as the expressivity of the model in successfully capturing a nonlinear, multiregion latent process.
>
>
> We comment below on specific weaknesses and suggestions brought up by the reviewer, detailing changes and additional analysis we have included. We hope that we have satisfactorily addressed some of the concerns and weaknesses brought up by the reviewer, and we apologize in advance that we were not able to fix all issues.
>
>
> **1. Main weakness: typos, incomplete legends and incorrect references affecting clarity of the paper.** We deeply apologize to the reviewer for the typos, incomplete legends, and incorrect figure references pointed out by the reviewer as the main weakness of our paper. We understand that these oversights deeply affect the clarity and accessibility of our work, as well as the overall impression of the reader of its quality. We also realize these presentation errors were a practical barrier to understanding and evaluating our experiments, and we again deeply apologize for this. **We have carefully edited and fixed** all typos and other mistakes, and we hope the paper is now clear, and thus can be better assessed.
>
>
> **2. Clarity of benchmarks table, adding standard deviation of results for benchmarks.** We apologize for the lack of clarity and have **modified the caption** to clarify - the table presents the mean square error, or MSE, of co-smoothed, held out neuron activity predictions. We use MSE since we apply the model to continuously valued calcium imaging traces, for which we use a Gaussian observation model. Regarding the need to include standard deviation of results, we thank the reviewer for this suggestion and apologize that we didn’t have a chance to update the table during the discussion time.
>
>
> **3. Single discrete switch in Lotka-Volterra simulation constrained in time is too simple to validate the model; clarity in the simulation figures.** We thank the reviewer for this insightful comment on the limitations of this simulation. **We have added results of an updated version of the simulation** which uses multiple switches , between 1 and 3 switches picked randomly on each trial. We **have also changed** the potential switching window to be uniformly sampled between timepoints 10-140 in a 150 timepoint long simulation. We present this new result **in Figure 2**. Additionally, we apologize for any confusion in the legend and traces in the original presentation of these results and **have updated these figures** to have clearer traces and legends.
>
>
> **4. Additional comparisons with other models on simulation experiments.** We thank the reviewer for pointing out this shortcoming in our presentation. **We include additional comparisons** on simulations in sections 4.1, 4.2, in *Figures 2 and 3* respectively. We compare with models the reviewers mentions, such as mr-LDS, mr-SLDS, and mr-rSLDS, visualize the latents recovered for each model, as well as quantify the relative performance.
>
>
> **5. Benefits of our model for future prediction and latent estimation over other models.**
> We thank the reviewer for calling attention to the fact that the relative benefit of our model over others was not sufficiently clear in the paper, despite achieving better performance on the presented benchmarks. We consider two aspects of this shortcoming, (a) the relative quantitative performance, and (b) the relative qualitative fits and scientific interpretation. We do 3 things to address both of these aspects. First, **we have introduced a modified objective in training the network**, which forces coupling between the generative dynamics and emissions networks during training (equation 14 in the updated paper) -- **resulting in higher accuracy of recovered latents, dynamics and messages** in our simulation experiments, as shown in **the updated Figures 2 and 3**.

---

> > ### Comment · Reviewer_LTdS · 2023-11-21
> >
> > Thanks for the authors' detailed comments. But I'm still curious about the response to my Weakness 4 and Question 3.

---

> > > ### Author Response · Authors · 2023-11-22
> > >
> > > Yes! We apologize for our oversight in not addressing these questions directly and thank the reviewer for their reminder.
> > >
> > >
> > > **Regarding question 4**, this is an important question, we apologize this point was left unclear and that we forgot to mention it in our response. As the reviewer suggests, the forward Euler is problematic because it will drift from the true solutions as a function of the step size, and this was indeed contributing to the issues with the earlier simulation. When generating the new, longer simulation **based on the reviewer’s feedback, we simulated the system with a much smaller step size**, dt=0.001, generating 75,000 steps for the numerical integration for stability, and then **downsampled the resulting timeseries**, with one sample for every 500 steps -- thus producing a final simulated time series of 150 timesteps for each trial. We found this improved the results in addition to the new objective, and as can be **seen in Figure 2**, we are able to recover the latent trajectories of the system nearly perfectly. We thank the reviewer again for this earlier comment, and **we have updated the text to clarify this issue**. Finally, we note that while this is still also a simplistic approach to integration of these equations, we decided not to resort to more sophisticated approaches, as the Lotka-Volterra equations themselves were not our object of study, and our intention was simply showing recovery of overall dynamics and communication in this simulation (as a model of oscillatory communication) -- and therefore may not require yet higher precision. We hope this is clear and satisfactory and thank the reviewer again for this helpful earlier question.
> > >
> > > **Regarding weakness 3** we apologize if this remained unclear in our responses. Table 1 shows the mean square error (MSE) on a particular measure of held out prediction, co-smoothing, popularized in the recent Neural Data Benchmarks to compare across neural data models (https://arxiv.org/abs/2109.04463). In co-smoothing, a subset of neurons (here 25%), are dropped out on held out test trials, and then predicted by the model using the remaining neurons in those trials. Thus, both trials and neurons are held out. This is a challenging target because it requires the model to learn to represent relationships amongst the neurons in its latent space so it can predict unseen neurons on unseen trials, as opposed to simply compressing and reproducing held out entire test trials. In Table 2 in the Appendix, we show results on a sweep of model hyperparameters for a related, even more challenging target, which we refer to as 1 step generative co-smoothing. In this form of co-smoothing, the model cannot generate predicted held out neurons directly from inferred latents, and has to pass the inferred latents through one timestep of the generative dynamics before passing through emissions, ie. $E_{q(x|y)} [p(y_t|f(x_{t-1}))]$. Thus, this target requires the generative dynamics to be accurate in addition to the latent inference, and this evaluation target is behind our updated training objective in Equation 14. **We have updated the captions for Tables 1 and 2 to make this clearer**, we also refer the reviewer to **section A.5 and Figure 9 in our updated Appendix** which gives more detail on co-smoothing and 1-step generative co-smoothing.
> > >
> > >
> > > Finally, we realize MSEs can be hard to compare and interpret, our choice of this metric was due to use of a Gaussian observation model for calcium imaging data, and the simplicity of computing MSE on saved predictions without loading the models - we will plan to include log likelihood instead as a more intuitive measure and apologize for not getting a chance to do this during the discussion period. (We also plan to include variance of each result as the reviewer suggested). To address the question of significance simply for the moment, **we added a plot (Figure 7 in the updated Appendix)** showing a Gaussian fit to the LDS model scores **along with the score and p-value for our model, which is highly significant** (mesoscope=2.209e-99, widefield=5.345e-4).

---

> > > > ### Comment · Reviewer_LTdS · 2023-11-22
> > > >
> > > > Thanks for these comments. I have raised my score from 3 to 5.

---

> > > > > ### Author Response · Authors · 2023-11-22
> > > > >
> > > > > We thank the reviewer deeply for their time and consideration

---

### Official Review · Reviewer_yiV5 · 2023-10-29

**Soundness:** 2 fair
**Presentation:** 3 good
**Contribution:** 2 fair
**Rating:** 6
**Confidence:** 4

**Summary:**

This paper proposes a switching nonlinear state space model to capture the nonlinear pattern in multi-region neural dynamics. The authors use a deep neural network-based approach, where they amortize the cost of inference by learning an inference neural network (Transformer in this paper), which predicts the mean and variance of the latent dynamics. In the experiments, they compare the proposed method with PCA and some widely used state space models in the neuroscience community by evaluating the prediction performances. They also apply the proposed method to multi-region calcium imaging data to explore the communication streams between regions.

**Strengths:**

Propose a switching nonlinear dynamics system by using the powerful expression of deep neural networks.

**Weaknesses:**

* The experiment part needs clarification. Section 4.2 simulates a multi-region decision-making task, but there is no comparison of MR-SDS with other methods such as LDS, SLDS, and rSLDS. Besides, the author mentions their method, MR-SDS, is able to learn the nonlinear dynamics in this complex case, but there needs to be a figure to visualize such latent dynamics. Similarly, section 4.3 simulates a multi-region system with three interacting and high-dimensional RNNs, but there needs to be a figure to support their claim.

* The comparisons with LDS, SLDS, and rSLDS are not enough. This paper only compares their prediction/reconstruction performance on calcium imaging data. It would be better to show the proposed method, MR-SDS, could capture more meaningful latent dynamics than other commonly used models, which may be more important than prediction performances in terms of neuroscience research. For example, could you provide some cases where MR-SDS captures some nonlinearity in the neural recordings while rSLDS cannot?

* No standard deviation (variance) in Table 1.

*  Many typos and wrong figure index/reference. E.g., in section 4.1 and the second line of page 6, it should be "Figure 2(c) shows ....". In section 4.1, there is no reference to Figure 2(d), etc.

**Questions:**

* Could SLDS / rSLDS give similar latent dynamics and message communications in Lokta-Voterra simulated data, mesoscope, and widefield calcium imaging data? Although SLDS is a linear model, it could capture the nonlinearity in latent space.

* Are there any insights as to why you choose Transformer instead of some simpler and faster deep neural networks like RNNs?

---

> ### Author Response · Authors · 2023-11-21
>
> We thank the reviewer for their time and evaluation of our paper, as well as for helpful suggestions and comments. We also appreciate their comment on the expressivity of our model in using deep neural networks to parameterize latent switching dynamics, and on our comparison in the paper with a number of widely used state space models in the neuroscience community.
>
>
> We comment below on specific points and suggestions brought up by the reviewer. We apologize in advance that we weren't able to address all issues.
>
>
> **1. Fixing typos and incorrect figure reference.** We deeply apologize for the typos and incorrect references pointed out by the reviewer, and we understand that these deeply affect the clarity and accessibility of our presentation, as well as the overall impression of the reader of the work. **We have carefully edited and fixed** all typos and other mistakes, and we hope the paper is now clear.
>
>
> **2. Clarification of experiments section, additional comparisons with other models on some of the simulations.** We thank the reviewer for pointing out this shortcoming in our presentation. First, **we have moved the key results into the main text** for section 4.2, to help support and visualize the result on this experiment. Second, **we include additional comparisons** on these simulations with other models the reviewers mentions, such as mr-LDS, mr-SLDS, and mr-rSLDS, in **Figures 2 and 3**. We visualize the latents recovered by mr-LDS and mr-rSLDS for the Lotka simulation, and for mr-rSLDS (the most advanced of these models) on the Double Well simulation for qualitative and quantitative assessment.
>
>
> **3. Standard deviation of results for benchmarks.** We thank the reviewer for this suggestion and apologize that we were unable to update the table in the discussion time.
>
>
>
> **4. Qualitative advantage of learned MR-SDS latents over rSLDS.** The reviewer makes a very important point - improved prediction performance is an important measure but may not be key to producing improved scientific interpretation, and thus it is important to demonstrate a qualitative advantage in latents learned by MR-SDS for neuroscience research. We agree that more work needs to be done in this direction and apologize that we were unable to provide additional analysis in the discussion time.
>
> **5. Choice of Transformer encoder vs RNN.** This is an important question - in developing the MR-SDS model we indeed tested an RNN based encoder (a stochastic, bi-drectional RNN, or Bi-RNN specifically, similar in spirit to the one used in LFADS), which did not perform as well as the Transformer, but we did not include those results in the submitted paper due to space constraints. **We have added an additional result in Table 1 showing performance of the Bi-RNN encoder** compared to the transformer. We note that because we constrain our use of the Transformer (or RNN) to the encoder alone, but use simple feed forward or MLP networks for the generative dynamics, we are not directly concerned with the complexity or interpretability of the encoder network, and thus can benefit from the high trainability and expressivity of the Transformer without complicating the generative model and resulting scientific interpretation. We have added additional detail on this last point to the discussion section.

---

> > ### Comment · Reviewer_yiV5 · 2023-11-21
> >
> > Thanks for conducting additional experiments and explanations. The paper has improved overall after fixing typos and incorrect figure references. I have upgraded the score from 3 to 6.

---

> > > ### Author Response · Authors · 2023-11-22
> > >
> > > We thank the reviewer deeply for their consideration!

---

### Official Review · Reviewer_rCyL · 2023-10-31

**Soundness:** 3 good
**Presentation:** 2 fair
**Contribution:** 2 fair
**Rating:** 6
**Confidence:** 4

**Summary:**

This paper presents a method for modeling interactions between brain regions in the context of a collection of nonlinear state space models. It aims to describe the neural dynamics both internal to a brain region and between regions. The authors provide plentiful comparisons to other similar methods, applied mostly to decision-making paradigms in neuroscience.

**Strengths:**

The intuition underlying the method is well described, and the math is clearly laid out. Quantifying the messages between regions in a nice addition beyond existing methods.

The method does outperform other (very similar) methods in the datasets tested here.

**Weaknesses:**

Some clarifications of the use cases / benefits of the method would be helpful. For instance why is 'switching dynamics enables modeling the dynamics in low-dimensional space using more persistent discrete states' a benefit? A clearer description of the goal is needed.
Overall the discussion was weak and read as a series of unjustified statements.

Some of the figures (e.g. Figure 5 B) were very low resolution, unlike other areas of the same figure. The caption would also be improved by defining terms ('accum' is different than 'accumulator'?, 'msgs'). Figure 5 has no C label.
Figures don't appear to be consistently formatted and the text is often too small to read. (except Figure 1, which is well done and easily understood even without a caption).

**Questions:**

The authors appear to have left in a note: 'MAKE SURE to say here that we also include in appendix A1 a result without the external input and show that we can still do inference but can’t generate'. (Why can't it generate?)

Could the authors comment on 'mr-rSLDS models required 10 latent dimensions per region (vs 3) to achieve comparable performance on the mesoscope dataset' and why more latent dimensions per region is a downside? Is it just about the computational load? Or is it important to model the latent dimensions with e.g. 3 dimensions because the brain itself 'uses' fewer than e.g. 10 dimensions? There is some trade off between summarization (fewer dimensions) and models that are structured more similarly to the higher-dimensional neural dynamics (more biological comparisons?).

Would this method be able to discern the number of interacting brain regions? Rather than take the regions are proscribed (3 regions, 16/8 regions in the experimental datasets).

---

> ### Author Response · Authors · 2023-11-21
>
> We thank the reviewer for their time in reading our paper and providing thoughtful and helpful comments, questions and suggestions.
>
>
> We especially appreciate the reviewer’s comments regarding the clarity of the intuition behind and mathematical development of our model presented in our paper, and regarding our novel contribution beyond existing methods in quantifying messages between regions. We also appreciate the reviewer’s recognition of our model’s outperformance of many other relevant methods on a number of datasets tested in the paper.
>
>
> We address the reviewer’s specific comments below, and apologize in advance that we weren’t able to address some of the specific concerns.
>
>
> **1. Clarifying the use case and benefits of the model, improving the discussion section.** We thank the reviewer for calling attention to this and regrettably agree that our discussion section was under-developed. **We have modified our discussion** to reflect the reviewer’s suggestions, focusing on the goals and benefits of our approach and clarifying and justifying our design decisions.
>
>
> **2. Disadvantage of many latent dimensions required in benchmarked piecewise linear models.** This is an important question brought up by the reviewer, and **we modify our discussion of this point** to better address and clarify. Briefly, the main problem with using piece-wise linear dynamics with many latent dimensions to capture low-dimensional nonlinear dynamics is interpretability. Koopman theory shows that it is possible to express any nonlinear dynamical system of finite dimensions as a linear system in a ‘lifted’, infinite dimensional space - so it is generally possible to use linear models of high dimensions to model a low dimensional nonlinear system. However, to visualize the dynamics of these systems in two or 3 dimensions, we have to use some dimensionality reduction method, such as PCA, to project those systems down, and in doing so we lose interpretability, since important features of the dynamics may live in the additional dimensions. A similar problem also exists with high-dimensional nonlinear approaches, such as LFADS. We agree with the reviewer’s insight that there is a trade off between models that can perform summarization, representing core dynamics of a system in fewer dimensions, and models that are more biologically plausible, and use high dimensional dynamics to capture the system. This trade off is actually the key motivation behind our high-dimensional RNN simulation, we regret we did not have a chance to flesh this out further.
>
>
> **3. Uncommented note on inference vs generation with external input in the switching Lotka Volterra simulation.** We apologize for our oversight of the visible comment that the reviewer pointed out.  Briefly, in the switching Lotka Volterra simulation, we provide an external switch cue to the model as an input, approximating an external stimulus cue or signal from an upstream region. Importantly, these switch times do not depend on the continuous latent state of the model, ie p(z|u,z), not p(z|x,z), similar to an input driven hidden markov model or HMM. Thus, while the model is able to infer the correct switching time without access to this external cue in inference mode, it is unable to *generate* correct switching times when run in forward generative mode without this cue, since z and x alone do not contain information about the switching times. We note that in the double well simulation, there is no such external switching cue, and the global switch depends directly on the continuous latent state x.
>
>
> **4. Figure formatting.** We deeply apologize for the inconsistent formatting, resolution, and captions and text size in some of the images that the reviewer points out, and **we have carefully edited and modified** all figures to reflect the reviewer’s comments.
>
>
> **5. Discerning the number of brain regions.** We thank the reviewer for this insightful question, as in some cases and species brain regions and their functional roles may be less well defined. Because our model constrains the emissions process so that each latent dimension uniquely and ‘privately’ maps to particular observation dimensions, we have to pre-specify which observations come from each region. It is in principle possible to additionally learn these mappings, **we add this as a future direction in our discussion**. To address this question partially, however, **we run a 16 region model on the widefield dataset**, for which we had collapsed the original 16 regions across hemispheres into 8 regions in order to simplify the presentation of results. Table 2 in the Appendix shows 16 regions provides only a marginal benefit over 8 on 1 step generated co-smoothing. This is a more challenging target than typical co-smoothing, which maps form inferred latents to observations, by requiring the model to co-smooth through a single timestep of generated dynamics.

---

> > ### Comment · Reviewer_rCyL · 2023-11-23
> >
> > Thanks to the authors for their additional clarifications, and the paper does read more clearly now. My overall score stands.

---

### Official Review · Reviewer_zWh9 · 2023-11-02

**Soundness:** 2 fair
**Presentation:** 2 fair
**Contribution:** 2 fair
**Rating:** 6
**Confidence:** 4

**Summary:**

This paper presents a novel state-space model for analyzing multi-region neural recordings. Briefly, the multi-region switching NLDS architecture assumes multiple, switching nonlinear latent dynamical systems per region, governed by a discrete state variable and continuous latent dynamics, whose transitions are all functions of the system input and previous states parameterized by neural nets. The emission model is likewise parameterized by an NN, while the latent state inference is performed by a Transformer that pools across regions (after local embeddings). Training is done via maximizing ELBO. The authors apply their method to 3 simulated systems, finding good performance compared to ground-truth latents and interpretable flow fields and inter-areal messages. Furthermore, the proposed method is benchmarked against a variety of previous single/multi-region switching/non-switching (non-)linear methods on two real datasets, with best performance on the co-smoothing metric. Lastly, they demonstrate how the method can be used to gain dataset-specific neuroscientific insights.

**Strengths:**

The proposed method is a technical tour-de-force, and represents a generalization of many existing algorithms (e.g., MR-SLDS). The 3 simulation experiments are quite nice, and demonstrate the variety of systems for which this method may be applicable, even in the case of a mismatched model (i.e., the multiregion RNN model). The benchmark experiments against existing methods is also appreciated. The paper is clearly and concisely written for the most part, especially the introductory pages and explanation of the method, though one gets the feeling that the authors ran out of time (or space) for the results half of the paper, and there are some “rough edges” throughout. Overall, it seems like a promising and well-developed method with somewhat natural assumptions for modeling multi-region interactions in the brain.

**Weaknesses:**

I have two main concerns, which I will outline in brief here, with specific comments/questions in the next section:

First, the method is powerful by design, leveraging multiple nonlinear DS per region. However, the flexibility comes with a series of issues, most prominently the choice of hyperparameter values, such as the dimensionality of the system and, critically, the number of switching states. In the simulation experiments, if I understand correctly, the model was given the ground-truth number of states, but how would the inferred systems look like when starting with a different number (e.g., for the switching Lotka-Volterra system)? This is further an issue with the real neural data, where one does not have access to the ground-truth, and it’s arguable whether there is such a number.

Second, is that the applications to real data is somewhat unconvincing for me, both in terms of the assumptions one makes regarding model architecture and hyperparameter values (related to above) and what actual insight was gained. Given the technical contribution in the method, I don’t think the authors should necessarily be penalized for attempting to apply it on real data. Nevertheless, I feel that the main claims regarding neuroscientific insight was not sufficiently evidenced.

Therefore, my recommendation is borderline reject, and I would be willing to update my score if the authors can conduct a few robustness experiments, as well as argue why some of the hyperparam decisions and model assumptions are not problematic in a neuroscience setting, and/or are acknowledged as limitations.

**Questions:**

- a major question is whether the existing results are robust to different choices of the number of switching states, latent dims, etc., especially in the simulation experiments but using an incorrect number, as well as how do results change when applied to experimental data. I leave up to the authors to decide how best to demonstrate this, but in general, for a practicing neuroscientist using such a method, I would expect somewhat robust/consistent results independent of hyperparam choices.
- The inferred systems from the real data (Figures 3 & 4) don’t look that great, and I’m generally not sure what I’m suppose to get from them in general. First, in Figure 3B, the flow fields in state all look like linear 1D flows. Is this meaningful? Could this not have been combined with the first state? Or what happens if one uses a larger number of states, are all the ones after state 1 simple linear-ish flows? And how would one interpret this either way?
- In general, the figures could use a bit more explanation, e.g., what are the main occupied states? Why is the same trajectory plotted in both states but different portions are highlighted in blue in 3B? How exactly should someone interpret 3C / 4D? Various plots don’t have labeled axes (e.g., 3D, 4B/C), I guess that’s time in ms?
- one assumption the model makes is that communication across regions are in the form of the latent variable, while in the brain this happens concretely and literally via spikes, which is more akin to the observed signal in this model (though here it’s calcium signals). How is the original assumption valid given this?
- in practice, how does one decide on the various hyperparameter values, such as the number of latent dimensions, switching states, etc.?
- Some more discussion regarding whether there is an advantage to using multiple discrete nonlinear DS instead of one bigger / flexible one is warranted. In particular, is the switching timescale realistic considering neurobiology, and what would the switching states represent?
- I’m not sure what to make of the comparison to PCA in the high-D RNN experiment: the inferred flow field look quite similar, and if anything the PCA one looks cleaner. Maybe some further expansion on what we should be looking for would be helpful? (Also what are the markers?)
- some small typos and such: page 6 line 2, “Figure 2a” should be 2c; line directly above figure 2 is a runaway latex comment; page 8 section 4.6 “Figure 5 shows a macro…” should be Figure 3; Supplemental text refers to Figure AX while the numbers continue from the main figures; Figure 3D missing caption label, and what is the dark brown suppose to be? Fixing these (and the figure issues above) would lift the overall quality of the paper.

---

> ### Author Response · Authors · 2023-11-21
>
> We thank the reviewer for their time and attention in reading our paper, for their very detailed and insightful comments and for their helpful suggestions and specific recommendations in improving our paper. We especially appreciate the reviewer’s kind remarks on the technical strength and novelty of our method in generalizing existing algorithms, characterizing it as promising and a tour-de-force. We also appreciate the comments regarding the quality of our simulation experiments and benchmarks, modeling assumptions, and the clarity of presentation of the introduction and method.
>
> Below, we address the comments the reviewer has made. We apologize that we weren’t able to directly address all of the concerns in the given time.
>
>
> **1. Small typos, ‘rough edges’, and additional explanation of figures.**
> We thank the reviewer for calling attention to specific small typos and places where the paper’s clarity could be improved. We regrettably agree that our results section was lacking in the clarity and polish of the introduction and model presentation, and **we have carefully edited and updated the paper** to reflect the reviewer’s suggestions.
>
>
> **2. Hyperparameter selection and tradeoffs of interpretability vs performance**
> We thank the reviewer for calling attention to this important issue. As the reviewer mentions, the expressivity and flexibility of a model with switching nonlinear dynamics and nonlinear emissions requires well-motivated choices of hyperparameters, specifically the choice of latent dimensions and number of discrete states. For the number of dimensions, we purposely choose 2 dimensions to represent each latent space across all simulations and datasets, reflecting the perspective that we desire to learn dynamics that are easy to visualize, ie can be visualized in 2d. This is so that we can avoid using a dimensionality reduction method such as PCA to only partially visualize and understand the dynamics of the model post-hoc, after it has been fit. However, it is important to understand the performance consequences of this design choice - therefore we **include in Table 2 in the Appendix** a sweep of hyperparameters, showing the single step generated co-smoothing performance across 2-5 dimensional latent models fit to the neural data, as well as for 1-3 states. This benchmark is a more challenging one, as it requires co-smoothing using a single time step of generated dynamics under the generative model (vs going directly from inferred latents to observations). These results show that while adding additional dimensions above 2 to the latent space improves performance, the 2 dimensional model compares well (and is easier to visualize and interpret). Similarly, adding an additional discrete latent state improves performance, but is comparable.
>
>
> **3. Robustness of the model’s scientific results across hyperparameter choices.** This is an important point, and we regret that we weren't able to address it fully. First, we note that hyperparameter choices in many models, including ‘simple’ ones, can lead to different scientific interpretations - for example, fitting a clustering model such as a hidden markov model (HMM) or K-means to data with different numbers of states can result in ‘lumping’ or ‘splitting’ of eg different neural activity or behavioral patterns, with potentially different scientific conclusions. There is a similar issue with the number of discrete states in our model, and **we have updated our discussion**, as per the reviewer’s suggestion. Second, **we include a simple analysis of robustness** of inferred message representations, by showing the across trial mean message norm profiles for the mesoscope data across models in the parameter sweep (Figure 6 in the Appendix), showing these look similar across models. We realize that more thought and analysis should go into assessing scientific robustness, we focus here on messages since we believe they are a key target for downstream scientific analysis.
>
>
> **4. Scientific interpretation and explanation of application to neural datasets.** We thank the reviewer for their thoughtful and nuanced comments on the matter of scientific interpretation of the neural datasets using the model. We agree that we have more work to do in this respect, and regret that we couldn’t include further analysis which is a work in progress. A small comment on the dynamics gradient fields plotted in Figures 4 and 5, those fields are only the local component of the dynamics acting on each of the regions, whereas the full induced dynamics field is a sum of the additional terms. To compute these, it is possible to average the empirical incoming gradients from all other regions on each point on the grid, we didn’t have a chance to include this regrettably - but the induced fields are likely much less linear.

---

> ### Comment · Reviewer_zWh9 · 2023-12-05
>
> Many thanks to the authors for their detailed responses to my questions and comments, and their acknowledgement of the current limitations and the lack of time to address them fully in revision.
>
> EDIT: I realize now that I would effectively be arguing for the paper's rejection given my current score, and I certainly don't feel confident enough to do so. Therefore, given my above comments, noting especially the authors' revision that improved low-level readability issues and acknowledgement of limitations, I updated my score from 5 to 6.

---

### Official Review · Reviewer_iayK · 2023-11-02

**Soundness:** 3 good
**Presentation:** 2 fair
**Contribution:** 2 fair
**Rating:** 6
**Confidence:** 4

**Summary:**

This paper presents a novel approach, Multi-Region Switching Dynamical Systems (MR-SDS), for modeling neural interactions across multiple brain regions. The proposed method models low dimensional nonlinear neural dynamics. With transformer encoders inferring the states, the model provides a more precise representation compared to existing models such as rSLDS and RNNs. The model is validated by three simulations and is applied to two multi-region neural datasets.

**Strengths:**

The authors extend existing models to make them nonlinear, while keeping key aspects of modeling such as low dimensional dynamics in the neural space. Switching dynamics are able to describe neural activity in a variety of behaviors. The model is able to perform well on held-out and co-smoothing tests.

**Weaknesses:**

1. The authors state that RNNs have higher dimensionality while solving such problems, and thus, it is difficult to interpret the inferred dynamics. However, MR-SDS has a transformer included which may be even more difficult to interpret. Indeed, while the authors show a higher accuracy, they do not show the interpretability at all; it is not clear how interpretable this model is, more so since the communication between different brain regions cannot be accurately recovered (e.g., Fig. 2D).
2. In Figure 2, what is rho? Additionally, the reconstruction of states and the reconstruction of states’ communication are actually not good, thus, it is hard to say the model reconstructs the latent states successfully.
3. Could the authors detail the parameters, such as alpha, beta, and so on, in section 4.1.
4. In section 4.3, it would be better to link the paragraph to the figure with the corresponding result, referring to which figure the authors are talking about. Additionally, I assume that the figure is Figure 6, but how do the authors find that MR-SDS embeds a richer representation than PCA? And why is a richer representation better here? Since the resulting PCs are orthogonal, it is unfair to compare them with the richness of the data representation.

Moreover, clarity is not as high as it should be. The authors would need to proofread the submission carefully and generate the figures professionally. Here are some examples that may need to be revised:
Section 4.1, last sentence: “MAKE SURE to say here that we also include in appendix A1 a result without the external input and show that we can still do inference but can’t generate” is not appropriate to show here.
Figure 2B needs a legend.

**Questions:**

See above.

---

> ### Author Response · Authors · 2023-11-21
>
> We thank the reviewer deeply for their time and attention in reading our paper, and for their very helpful comments. We also appreciate their characterization of our work as novel and for noting that our model extends and generalizes earlier models and performs well on benchmarks. Below we respond to specific weaknesses pointed out by the reviewer, and detail changes to the paper and new analysis that addresses these.
>
>
>
> **1. Clarity could be improved.** We thank the reviewer for pointing out specific proofreading errors in the paper, as well as issues with the figures such missing legends in subfigures. **We have carefully edited the paper** to remove all errors, correct omissions, and ensure clarity of the figures. We have also **added the specific parameters** used for the simulation parameters (alpha, beta, gamma, delta) inline in section 4.1 as the reviewer suggests. Finally, as per the reviewer’s suggestion, we **have moved core result figures** to the main text following the paragraphs for section 4.2, with additional figures in the appendix. We hope the reviewer finds these improvements satisfactory.
>
>
>
> **2. Interpretability of transformer inference model in MR-SDS vs RNN-based generative models.** The reviewer rightly points out that while we mention in the paper that high-dimensional RNN generative models can suffer from a lack of interpretable dynamics, our model uses a transformer, which is also hard (and maybe harder) to interpret. We would like to emphasize however that we constrain our use of the transformer to the *inference* step alone, and use simple feed forward or MLP networks to parameterize the nonlinear dynamics functions in the generative model. Thus, while the transformer inference network is indeed complex to interpret, our model does not require interpreting it to analyze the dynamics of the system. On the other hand, the generative model we use is easy to interpret because it produces low-dimensional dynamics gradient fields which are designed for easy visualization in 2 dimensions. This property contrasts with high-dimensional RNN models, (such as LFADS) in which the generative dynamics are produced in a very high-dimensional space, often in the hundreds or higher, thus requiring dimensionality reduction methods such as PCA to visualize dynamics, and then only partially (as important aspects of dynamics may exist beyond the first 2-3 PCs). **We have updated our discussion section** to make this issue clearer.
>
>
>
> **3. Accuracy of dynamics and cross-region message reconstruction in Lotka Volterra simulation and interpretability of the model.** We thank the reviewer for calling attention to the fact that recovered dynamics gradient fields and communications profiles were imperfect despite high performance of the model, harming interpretability of the model. To address the issue of generative dynamics and messages accuracy, **we have introduced a modified objective** in training the network, which forces coupling between the generative dynamics and emissions networks during training (**equation 14** in the updated paper) -- **resulting in more accurate recovered dynamics and messages** as seen in our **updated Figure 2**. We also apologize for the omission in the original figure - rho is a correlation coefficient computed between true and inferred messages. We now cite the updated correlation coefficient in the figure caption.
>
>
>
> **4. MR-SDS embeds a richer representation of coupled high-dimensional RNNs vs PCA.**  We apologize for the lack of clarity on this point, made worse by the unclear figure and reference. This is a comparison of the explained variance in the simulated observations reconstructed under PCA vs MR-SDS embeddings. The author rightly points out that PCA dimensions have an orthogonality constraint that complicates their comparison with the latent dimensions of our model. We add to this, that PCA embeds data linearly, an additional constraint, while our model embeds data nonlinearly. In this sense, it is an ‘unfair’ comparison, as PCA is clearly a less expressive method. Our point in this comparison is simply to compare a practical method used by many researchers in the field to visualize and understand the dynamics of high-dimensional network models (such as LFADS), with our method, which directly produces low-dimensional dynamics fields that explain the data. With access to the full *true* network dynamics (which a researcher would not have in the case of true data, only in a simulation), a PCA embedding of the full, ground-truth dynamics still captures less variance in the observations than the low-dimensional dynamics inferred by MR-SDS, and thus would offer a less complete description of the data and dynamics. This makes the point that MR-SDS does a good job of modeling a high-dimensional system, despite model mismatch.

---

### Meta-Review · Area_Chair_FiqC · 2023-12-10

**Metareview:**

This paper presents a approach called Multi-Region Switching Dynamical Systems (MR-SDS) for modelling neural interactions across multiple brain regions. The proposed method models low dimensional nonlinear neural dynamics. With transformer encoders inferring the states, the model provides a more precise representation compared to existing models such as rSLDS and RNNs. The model is validated by three simulations and is applied to two multi-region neural datasets.

The work builds on a number of works using dynamical system with latents for neural data.

Reviewers acknowledge merit in this work and do not report major flaws, yet do not fully champion the paper.
The opinion is however overall positive and the use of controlled experiments with simulations is appreciated.

**Justification For Why Not Higher Score:**

The impact is possibly limited for the ICLR community.

**Justification For Why Not Lower Score:**

Paper with overall positive reviews and no major flaw.

---

### Decision · Program_Chairs · 2024-01-16

Accept (poster)